# Deep Symbolic Superoptimization without Human Knowledge

**Hui Shi**[1*], **Yang Zhang**[2*], **Xinyun Chen**[3], **Yuandong Tian**[4], **Jishen Zhao**[1]
UC San Diego[1], MIT-IBM Watson AI Lab[2], UC Berkeley[3], Facebook AI Research[4]
{*hshi, jzhao*}@*ucsd.edu, yang.zhang2@ibm.com, xinyun.chen@berkeley.edu, yuandong@fb.com*

## Abstract

Deep symbolic superoptimization refers to the task of applying deep learning methods to simplify symbolic expressions. Existing approaches either perform supervised training on human-constructed datasets that define equivalent expression pairs, or apply reinforcement learning with human-defined equivalent transformation actions. In short, almost all existing methods rely on human knowledge to define equivalence, which suffers from large labeling cost and learning bias. We thus propose HISS, a reinforcement learning framework for symbolic superoptimization that keeps humans outside the loop. HISS introduces a tree-LSTM encoder-decoder network with attention to ensure tractable learning. Our experiments show that HISS can discover more simplification rules than existing human-dependent methods, and can learn meaningful embeddings for symbolic expressions, which are indicative of equivalence.[1]

## 1 Introduction

Superoptimization refers to the task of simplifying and optimizing over a set of machine instructions, or code (Massalin, 1987; Schkufza et al., 2013), which is a fundamental problem in computer science. As an important direction in superoptimization, symbolic expression simplification, or symbolic superoptimization, aims at transforming symbolic expression to a simpler form in an effective way, so as to avoid unnecessary computations. Symbolic superoptimization is an important component in compilers, *e.g.* LLVM and Halide, and it also has a wide application in mathematical engines including Wolfram[2], Matlab, and Sympy.

Over recent years, applying deep learning methods to address symbolic superoptimization has attracted great attention. Despite their variety, existing algorithms can be roughly divided into two categories. The first category is supervised learning, *i.e.* to learn a mapping between the input expressions and the output simplified expressions from a large number of human-constructed expression pairs (Arabshahi et al., 2018; Zaremba & Sutskever, 2014). Such methods rely heavily on a human-constructed dataset, which is time- and labor-consuming. What is worse, such systems are highly susceptible to bias, because it is generally very hard to define a minimum and comprehensive axiom set for training. It is highly possible that some forms of equivalence are not covered in the training set, and fail to be recognized by the model. In order to remove the dependence on human annotations, the second category of methods leverages reinforcement learning to autonomously discover simplifying equivalence (Chen et al., 2018). However, to make the action space tractable, such systems still rely on a set of equivalent transformation actions defined by human beings, which again suffers from the labeling cost and learning bias.

In short, the existing neural symbolic superoptimization algorithms all require human input to define equivalences. It would have benefited from improved efficiency and better simplification if there were algorithms independent of human knowledge. In fact, symbolic superoptimization should have been a task that naturally keeps human outside the loop, because it directly operates on machine code, whose consumers and evaluators are machines, not humans.

---

*Authors contributed equally to this paper.

[1]The code is available at `https://github.com/shihui2010/symbolic_simplifier`.

[2]`https://www.wolframalpha.com/`

Therefore, we propose Human-Independent Symbolic Superoptimization (HISS), a reinforcement learning framework for symbolic superoptimization that is completely independent of human knowledge. Instead of using human-defined equivalence, HISS adopts a set of unsupervised techniques to maintain the tractability of action space. First, HISS introduces a tree-LSTM encoder-decoder architecture with attention to ensure that its exploration is confined within the set syntactically correct expressions. Second, the process of generating a simplified expression is broken into two stages. The first stage selects a sub-expression that can be simplified and the second stage simplifies the sub-expression. We performed a set of evaluations on artificially generated expressions as well as a publicly available code dataset, called the Halide dataset (Chen & Tian, 2018), and show that HISS can achieve competitive performance. We also find out that HISS can automatically discover simplification rules that are not included in the human predefined rules in the Halide dataset.

## 2    RELATED WORK

Superoptimization origins from 1987 with the first design of Massalin (1987). With the probabilistic testing to reduce the testing cost, the brute force searching is aided with a pruning strategy to avoid searching sub-spaces that contains pieces of code that have known shorter alternatives. Due to the explosive searching space for exhaustive searching, the capability of the first superoptimizer is limited to only very short programs. More than a decades later, Joshi et al. (2002) presented Denali, which splits the superoptimization problem into two phases to expand the capability to optimize longer programs. STOKE (Schkufza et al., 2013) follows the two phases but sacrifices the completeness for efficiency in the second phase.

Recent attempts to improve superoptimization are categorized into two fields: exploring transformation rules and accelerating trajectory searching. Searching the rules are similar to the problem of superoptimization on limited size program, but targeting more on the comprehensiveness of the rules. Buchwald (2015) exhaustively enumerates all possible expressions given the syntax and checks the equivalence of pairs of expressions by SMT solver. A similar method with an adaption of the SMT solver to reuse the previous result is proposed by Jangda & Yorsh (2017). On the other hand, deep neural networks are trained to guide the trajectory searching (Cai et al., 2018; Chen & Tian, 2018).

Considering transformation rule discovery as a limited space superoptimization, the large action space and sparse reward are the main challenges for using neural networks. Special neural generator structures are proposed for decoding valid symbolic programs, which leverage the syntax constraints to reduce the searching space as well as learn the reasoning of operations, and are gaining popularity in program synthesis (Parisotto et al., 2016; Zhong et al., 2017; Bunel et al., 2018), program translation (Chen et al., 2018; Drissi et al., 2018), and other code generation tasks (Ling et al., 2016; Alvarez-Melis & Jaakkola, 2016). Among the symbolic expression decoders, the family of tree structure RNNs (Parisotto et al., 2016; Drissi et al., 2018; Alvarez-Melis & Jaakkola, 2016; Chen et al., 2018) are more flexible than template-based predictors (Ling et al., 2016; Zhong et al., 2017).

## 3    THE HISS ARCHITECTURE

In this section, we will detail our proposed HISS architecture. We will first introduce a few notations. $\mathcal{T}$ denotes a tree; $\boldsymbol{a}$ denotes a vector, and $\boldsymbol{A}$ denotes a matrix. We introduce an $\text{LSTM}(\cdot)$ function that summarizes standard one-step LSTM operation as

$$[\boldsymbol{h}_t, \boldsymbol{c}_t] = \text{LSTM}(\boldsymbol{x}_t, \boldsymbol{h}_{t-1}, \boldsymbol{c}_{t-1}), \tag{1}$$

where $\boldsymbol{h}_t$, $\boldsymbol{c}_t$ and $\boldsymbol{x}_t$ denote the output, cell and input at time $t$ of a standard LSTM respectively.

### 3.1    FRAMEWORK OVERVIEW

Our problem can be formulated as follows. Given a symbolic expression $\mathcal{T}_I$, represented in the *expression tree form*, our goal is to find a simplified expression $\mathcal{T}_O$, such that 1) the two expressions are equivalent, and 2) $\mathcal{T}_O$ contains a smaller number of nodes than $\mathcal{T}_I$.

It is important to write the symbolic expressions in their expression tree form, rather than strings because HISS will be operating on tree structures. An expression tree assigns a node for each operation or variable. Each non-terminal node represents an operation, and each terminal node, or leaf node,

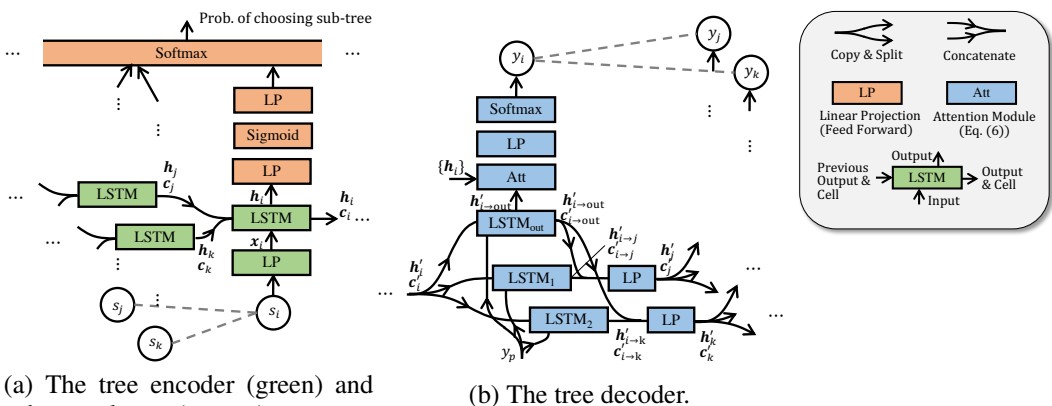

(a) The tree encoder (green) and subtree selector (orange).

(b) The tree decoder.

Figure 1: The HISS architecture, illustrated on a three-node binary subtree, where $i$ is the parent of $j$, $k$, and $p$ is the parent of $i$.

represents a variable or a constant. The arguments of operation are represented as the descendant subtrees of the corresponding node. Compared to string representation, tree representation naturally ensures any randomly generated expression in its form is syntactically correct. It also makes working with subexpressions easier: simply by working with subtrees.

HISS approaches the problem using the reinforcement learning framework, where the action of generating simplified expressions are divided into two consecutive actions. The first action is to pick a subexpression (or subtree) that can be simplified, and the second action generates the simplified expression for the selected subexpression.

Accordingly, HISS contains three modules. The first module is a **tree encoder**, which computes an embedding for each subtree (including the entire tree) of the input expression. The embeddings are useful for picking subtree for simplification as well as simplifying a subtree. The second module is a **subtree selector**, which selects a subtree for simplification. The third module is a **tree decoder** with an attention mechanism, which generates a simplified expression based on the input subtree embedding. The subsequent subsections will introduce each module respectively.

### 3.2 THE TREE ENCODER

The tree encoder generates embedding for every subtree of the input expression. We apply the $N$-ary Tree LSTM as proposed in Tai et al. (2015), where $N$ represents the maximum number of arguments that an operation has. It is important to note that although different operations have a different number of arguments, for structural uniformity, we assume that all operations have $N$ arguments, with the excessive arguments being a NULL symbol.

The tree encoder consists of two layers. The first layer is called the embedding layer, which is a fully-connected layer that converts the one-hot representation of each input symbol to an embedding. The second layer is the tree LSTM layer, which is almost the same as the regular LSTM, except that the cell information now flows from the children nodes to their parent node. Formally, denote $c_i$, $h_i$, and $x_i$ as the cell, output and input of node $i$ respectively. Then the tree LSTM encoder performs the following information

$$[\boldsymbol{h}_i, \boldsymbol{c}_i] = \text{LSTM}\left(\boldsymbol{x}_i, \bigcup_{j\in\mathbb{D}(i)} \boldsymbol{h}_j, \bigcup_{j\in\mathbb{D}(i)} \boldsymbol{c}_j\right), \tag{2}$$

where $\mathbb{D}(i)$ denotes the set of children of node $i$. Fig. 1(a) plots the architecture of the tree LSTM encoder (in green). Since each node fuses the information from its children, which again fuse the information from their own children, it is easy to see that the output $\boldsymbol{h}_i$ summarizes the information of the entire subtree led by node $i$, and thus can be regarded as an embedding for this subtree.

### 3.3 THE SUBTREE SELECTOR

The subtree selector performs the first action to select a subtree for simplification. It takes the output of the tree encoder, $\{\boldsymbol{h}_i\}$, as its input, and produces the probability with which each subtree is

selected. It consists of two feed-forward layers followed by a softmax layer across all the nodes in the input tree. Figure 1(a) shows the architecture of the subtree selector (in orange).

## 3.4 THE TREE DECODER

Once a subtree has been selected, and suppose the root node of the selected subtree is node $i$, the output of the encoder at node $i$, $\boldsymbol{h}_i$, is then fed into the tree decoder, which generates a simplified version of the subtree. The tree decoder can be regarded as the inverse process of the tree encoder: the latter fuses information from the children to the parents, whereas the former unrolls the information from parents down to the entire $N$-ary tree. When the parent node outputs a non-operation symbol, the expansion of this branch terminates and no child is further decoded.

The tree decoder adopts a novel LSTM architecture with attention, which, compared with the attention LSTM proposed by Chen et al. (2018), is more parameter- and computationally-efficient. It consists of two layers. The first layer is a tree LSTM layer, and the second layer is the symbol generation layer with attention. Figure 1(b) illustrates the decoder structure. The decoder shares the same vocabulary with the encoder, and the embedding layer of the decoder shares the parameters with the embedding layer in the encoder.

**Tree LSTM Layer.** The tree LSTM in the decoder needs to accomplish two tasks. First, it needs to extract the information for generating the output for the current node. Second, it needs to split and pass on the information to its children. To better control the information flow, we introduce two tracks of LSTMs for the two different tasks. Formally, denote $[\boldsymbol{h}_i', \boldsymbol{c}_i']$ as the output and cell of node $i$, and assume $[j_1, \cdots, j_N]$ are children nodes of $i$. Also denote $\boldsymbol{y}_p$ as the decoder output for node $p$, which is the parent node of node $i$ (If node $i$ is already the root node of the selected subtree, then $\boldsymbol{y}_p$ becomes a special start token). Then the first LSTM track extracts the information that generates the current output:

$$[\boldsymbol{h}_{i\to\text{out}}', \boldsymbol{c}_{i\to\text{out}}'] = \text{LSTM}_{\text{out}}(\boldsymbol{y}_p, \boldsymbol{h}_i', \boldsymbol{c}_i'). \tag{3}$$

The second LSTM track splits and passes on the information to the children, *i.e.* $\forall n \in \{1, \cdots, N\}$

$$[\boldsymbol{h}_{i\to j_n}', \boldsymbol{c}_{i\to j_n}'] = \text{LSTM}_n(\boldsymbol{y}_p, \boldsymbol{h}_i', \boldsymbol{c}_i'). \tag{4}$$

Notice that we have appended a subscript to the $\text{LSTM}(\cdot)$ to emphasize that LSTM functions with different subscripts do not share parameters. Finally, the LSTM information for a specific children is derived by linearly projecting the output track and that specific children track:

$$\boldsymbol{h}_{j_n}' = \boldsymbol{W}_h[\boldsymbol{h}_{j\to\text{out}}', \boldsymbol{h}_{i\to j_n}'] + \boldsymbol{b}_h, \quad \boldsymbol{c}_{j_n}' = \boldsymbol{W}_c[\boldsymbol{c}_{j\to\text{out}}', \boldsymbol{c}_{i\to j_n}'] + \boldsymbol{b}_c. \tag{5}$$

We find that this linear projection is useful for adding additional dependencies between the parent output and the descendants, so that the generated expression is more coherent.

**Symbol Generation Layer with Attention.** The symbol generation layer takes the output track produced by the previous tree LSTM layer, $\boldsymbol{h}_{i\to\text{out}}'$, as input, and outputs the probability distribution of generating different output symbols for the current node. It adopts an attention mechanism (Bahdanau et al., 2014) to attend to the relevant part in the encoder, so that the input and output expressions have better correspondence. Formally, when generating the output for decoder node $i$, the attention weight on encoder node $j$ is computed from $\boldsymbol{h}_{i\to\text{out}}'$ and $\boldsymbol{h}_j$ as follows:

$$\begin{aligned} e_i(j) &= \boldsymbol{v}^T \tanh(\boldsymbol{W}_d \boldsymbol{h}_{i\to\text{out}}' + \boldsymbol{W}_e \boldsymbol{h}_j + \boldsymbol{b}_a), \\ [a_i(1), \cdots, a_i(J)] &= \text{softmax}([e_i(1), \cdots, e_i(J)]), \end{aligned} \tag{6}$$

where $J$ is the total number of input nodes at the encoder. Finally, the probability of symbol generation at node $i$, denoted as $\boldsymbol{p}_i$, is computed by passing into a linear projection layer $\boldsymbol{h}_{i\to\text{out}}'$ and an attention context vector $\boldsymbol{c}_i$, which is a linear combination of the encoder embeddings with the attention weights, *i.e.*

$$\boldsymbol{p}_i = \boldsymbol{W}_o[\boldsymbol{h}_{i\to\text{out}}'; \boldsymbol{c}_i] + \boldsymbol{b}_o, \quad \text{where } \boldsymbol{c}_i = \sum_{j=1}^{J} a_i(j)\boldsymbol{h}_j. \tag{7}$$

## 4 LEARNING WITH HISS

In this section, we will elaborate on the training and inference schemes of HISS. In particular, we will introduce several mechanisms to improve the exploration efficiency of HISS.

## 4.1 TRAINING

We apply the standard REINFORCE framework (Williams, 1992) for training, where the reward function is given by

$$R(\mathcal{T}_I, \mathcal{T}_O) = \gamma^{\text{card}(\mathcal{T}_O)} \text{ if } \mathcal{T}_I \equiv \mathcal{T}_O, \quad -\beta\gamma^{\text{card}(\mathcal{T}_O)} \text{ otherwise,} \tag{8}$$

where '$\equiv$' denotes that the two expressions are equivalent; $\text{card}(\cdot)$ denotes the number of nodes in the tree expression, or the length of the expression. $\beta$ is a hyperparameter that depicts the penalty of not producing an equivalent expression. This reward prioritizes equivalence, and given equivalence, favors shorter expressions. We applied a probabilistic testing scheme to determine equivalence as proposed in Massalin (1987). For each input, multiple outputs are decoded via beam search, on each of which a reward is evaluated. We introduce the following mechanisms to maintain the efficiency and stability of training.

**Curriculum Learning.** Since generating the simplified expression is divided into two actions, subtree selection and subtree simplification, directly learning both can lead to very inefficient exploration. Instead, we introduce a two-stage curriculum. The first stage trains only the encoder and decoder on very short expressions (maximum depth less than four). The subtree selector is not trained. Instead, we always feed the entire tree to the decoder for simplification. The second stage trains all the modules on longer expressions.

**Subtree Embedding Similarity.** In order to guide the encoder to learning meaning embeddings, we introduce an additional $\ell_2$ loss to enforce that the equivalent expressions have similar encoder embeddings, *i.e.* similar $\boldsymbol{h}_i$s . Specifically, for each input expressions $\mathcal{T}_I$, we decode a set of generated expressions $\mathbb{S} = \{\mathcal{T}_O\}$ with beam search, and obtain their embeddings $\{\boldsymbol{h}(\mathcal{T}_O)\}$ by feeding them back into the encoder (here we add an argument to $\boldsymbol{h}$ to emphasize that the embedding is a function of input expression). Then the $\ell_2$ loss is expressed as follows

$$L = \frac{1}{|\mathbb{S}|} \sum_{\mathcal{T}_O \in \mathbb{S}} \|\boldsymbol{h}(\mathcal{T}_O) - \boldsymbol{h}(\mathcal{T}_I)\|_2^2 \cdot (-1)^{\mathbb{1}[\mathcal{T}_I \not\equiv \mathcal{T}_O]}, \tag{9}$$

where $\mathbb{1}[\cdot]$ denotes the indicator function, which equals one if the statement in its argument is true, and zero otherwise. Note that this $\ell_2$ applies to the encoder only, and can be optimized by regular gradient descent methods. REINFORCE is not needed.

## 4.2 INFERENCE

Similar to training, the inference is performed by decoding multiple outputs via beam search and finding the best result as the final output. In order to accelerate the inference process, we introduce an offline procedure. During the first stage of the curriculum training, *i.e.* training on very short expressions, all the simplified equivalence rules discovered are logged. During inference, if the subtree to be fed into the decoder has an exact match in the log, we will apply the logged simplified equivalence directly, rather than redoing the entire decoding process.

## 5 EXPERIMENTS

We performed two experiments. The first experiment compares HISS with human-independent naive search algorithms. The second experiment compares HISS with existing human-dependent state-of-the-art systems on benchmark datasets. Throughout all the experiments, we use the same hyperparameter setting. Additional details on the hyperparameters and how the hyperparameters are determined can be found in appendix A.3. Besides the experiments discussed in this section, some additional experiment results are reported in appendix B and a set of ablation studies are introduced in appendix C.

## 5.1 COMPARING WITH HUMAN-INDEPENDENT METHODS

Since there are no existing human-independent methods specifically for symbolic superoptimization, we compare several search algorithms. Due to the search complexity, the evaluation cannot be performed on very long expressions. Therefore, this experiment is performed on the traverse equivalence dataset.

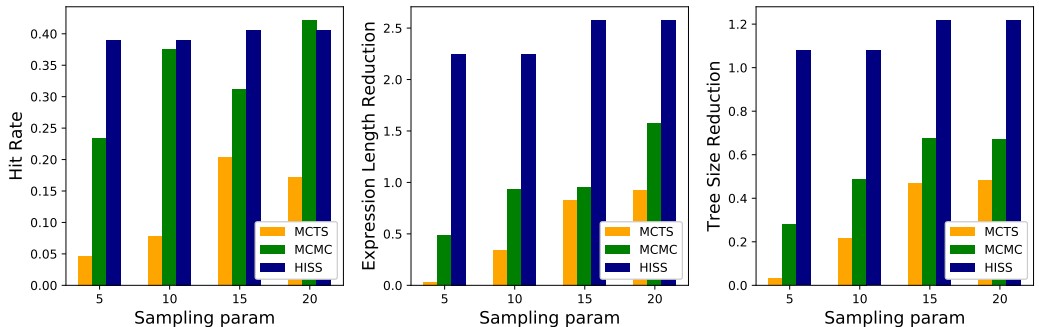

Figure 2: Comparison with human-independent methods in terms of hit rate (left), expression length reduction (middle), tree size reduction (right), on Traverse Equivalence dataset.

**Traverse Equivalence Dataset** As a complement to the Halide dataset, we build a dataset by traversing all the possible expressions with maximum depth four that consist of operations and symbols drawn from the Halide vocabulary in table 3. Among these expressions, we use the FingerPrint method (Jia et al., 2019) to test if they can be simplified, and discard those that cannot. From the remaining expressions, we sample 900 expressions as the training set, 300 as the validation set and 300 as the test set. The advantage of this dataset is that it is not built from human-predefined equivalence relation. However, the disadvantage of this dataset is that it does not contain expressions with a maximum depth greater than four, limited by the complexity of the FingerPrint method. Additional details of our Traverse Equivalence dataset can be found in appendix A.1.

**Training** Since HISS does not operate on very long expressions, it is only trained with stage-one in curriculum learning (the one with no subtree selection). Additional details regarding training can be found in appendix A.4.

**Baselines** Two baseline searching methods are compared: Monte Carlo Tree Search (MCTS) (Bertsekas, 1995) and Markov Chain Monte Carlo (MCMC) (Schkufza et al., 2013). MCTS decides the expression tree from root to leaves, choosing one symbol from Halide vocabulary for each node, and adopts Upper Confidence Bound (Kocsis & Szepesvári, 2006) for balancing exploration and exploitation. Similar to Schkufza et al. (2013), MCMC takes one of four transformations: 1) replace an operator by another random operator, and generate or discard operands if two operator takes a different number of operands. 2) replace a variable/constant with another random variable/constant. 3) replace a subexpression with a random single variable/constant. 4) replace a variable/constant with a random expression. The probability distribution of taking the transformation is defined as the same as in Schkufza et al. (2013).

**Metrics.** Three metrics are introduced: 1) *hit rate*, defined as percentage of expressions that the model successfully found an equivalence given the computation budget parameter; 2) *expression length reduction*, defined as reduction in the total number of tokens; and 3) *tree size reduction*, defined as reduction in the number of nodes in the expression tree.

**Results.** The performance comparison of three models is shown in figure 2. The sampling parameter in the horizontal axis refers to the beam size for HISS , the max trials budget for MCTS for each token decoded, and the sampling budget for MCMC. These quantities equivalently define the number of

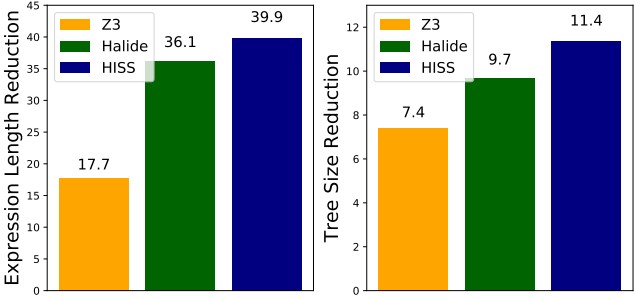

Figure 3: Comparison with human-dependent methods in terms of expression length reduction (left), and tree size reduction (right), on the Halide dataset.

Table 1: Example simplification rules learned by HISS (left) and their corresponding rules listed in Halide (right). If the learned rule has no corresponding preset rule in Halide, the row is left blank.

| HISS | | Halide | |
|---|---|---|---|
| Input | Output | Input | Output |
| (c0+v0) -(v0+c0) | 0 | | |
| (c0/v0)*v0 | c0 | (y*x)/x | y |
| c0-c0 | 0 | x-x | 0 |
| $c0 \leq c0$ | true | $x < x$ | false |
| c0 == c0 | true | x == x | true |
| $\min(c0, v0) \leq v0$ | true | $x < \min(x, y)$ | false |
| $\min(c0 + v0, c1) \leq (c0 + v0)$ | true | | |
| $\min(c0, (\min(v0, c1) + c2) + c1) \leq ((c1 + c2 + c1)$ | true | | |
| (c0 + v0) - c0 | v0 | (x + y) - x | y |
| max(c0, c0) | c0 | max(x, x) | x |
| $\min(c0, v0 + c1) \leq (c1 + v0)$ | true | | |
| $c0 \leq \max(v0, c0)$ | true | $\max(y, x) < x$ | false |
| c0 / c0 | 1 | x / x | 1 |
| $\min(c0 * v0, c1) \leq (c0 * v0)$ | true | | |
| c0 + (v0 - c0) | v0 | x + (y - x) | y |
| $\min(\min(c0, v0), c1 - c2) \leq v0$ | true | | |
| $\min(\min(c0, v0), v1) \leq v0$ | true | | |

search attempts per token. As can be seen, HISS is significantly more powerful in finding the simpler equivalent than MCTS and MCMC. MCMC performs almost equally well as HISS in terms of Hit Rate, and both of them far outperform MCTS. However, both MCTS and HISS adopt top-down decoding in the huge decoding space, while MCMC starts with the input expression and applies local transformations, which makes it much easier to find an equivalence. Also, MCMC achieves much worse average length reduction and average tree size reduction than HISS.

## 5.2 COMPARING WITH HUMAN-DEPENDENT METHODS

In this section, we compare HISS with existing human-dependent state-of-the-art methods on the Halide dataset.

**Halide Dataset** The Halide dataset (Chen & Tian, 2018) is the benchmark dataset for symbolic expression simplification. It consists of equivalent expression pairs constructed from human predefined rules. Since HISS is an unsupervised method, we only use the longer expression from each pair for training. Additional details of the Halide dataset can be found in appendix A.2.

**Training & Inference** In this experiment, HISS is trained with both stages of curriculum learning. The first stage is trained on the moderately short expressions from the Traverse Equivalence dataset, whose configuration follows that in section 5.1. The second stage is trained on the Halide training set in an unsupervised manner. More details can be found in appendix A.4.

Since HISS only simplifies one subtree at a time, while the actual simplification usually requires sequentially simplifying multiple subtrees, we iteratively invoke the HISS procedure for both training and inference, as in Chen & Tian (2018). Table 2 shows an example of the iterative process. The iterations terminate when 1) the number of iterations reaches 20; 2) the simplification output becomes a single node; or 3) the subtree selector assigns small scores (below 0.05) to all subtrees.

**Baselines** Two baselines are included: 1) *Halide* (Ragan-Kelley et al., 2013), which applies Halide predefined rules; 2) *Z3*, the simplification function in Z3, a high-performance theorem prover developed by De Moura & Bjørner (2008), to perform transformations using the Halide predefined rules. It is worth mentioning that both baselines have access to the Halide predefined rules that are used to construct the dataset, which gives them an advantage over HISS.

**Metrics** Expression length reduction and tree size reduction are applied as the metrics.

**Results** Figure 3 shows the performance of HISS compared with the baselines. As can be seen, HISS outperforms both Halide and Z3 in both metrics. This result is quite non-trivial because the Halide dataset is built exclusively from the Halide's ruleset, to which both baseline algorithms have access. Therefore, this result implies that even for expressions that are specifically designed to be simplified by a set of predefined rules, they can still be further simplified by rules that are outside of the set.

Table 2: Simplification process of HISS (upper) and Halide (lower) on Eq. (10). The subtrees selected for simplification in the next iteration are marked with box unless the entire tree is selected.

| Step | HISS |
|------|------|
| 0 | $(((((144 - (v0*72))/2)*2) + 4) \leq \boxed{((150 - (v0*72))/4)*4}$ |
| 1 | $\boxed{(((((144 - (v0*72))/2)*2) + 4)} \leq (150 - (v0*72)))$ |
| 2 | $(((144 - (v0*72)) + 4) \leq (150 - (v0*72))$ |
| 3 | $((144 - (v0*72)) \leq (146 - (v0*72)))$ |
| 4 | $\boxed{(((v0*72) -(v0*72))} \leq 2$ |
| 5 | $0 \leq 2$ |
| 6 | True |

| Step | Halide |
|------|--------|
| 0 | $(((((144 - (v0*72))/2)*2) + 4) \leq \boxed{((150 - (v0*72))/4)*4}$ |
| 1 | $((((72 - (v0*36))*2) + 4) \leq (((150 - (v0*72))/4)*4))$ |
| 2 | $((((72 - (v0*36))*2) + 4) \leq ((37.5 - (v0*18))*4))$ |
| 3 | $!((37.5 - (v0*18))*4)) < ((((72 - (v0*36))*2) + 4)$ |

## 5.3 SIMPLIFICATION PROCESS ANALYSIS

In this subsection, we will provide an in-depth analysis of the simplification process of HISS and Halide, which can explain why human-predefined rules can be insufficient, and how HISS can exceed the limit of human-predefined rules.

**Example Simplification Rules** To start with, table 1 lists some example simplification rules learned by HISS . For each example rule, the corresponding human-predefined rule in Halide is also listed if available. There are two observations. First, HISS is able to learn the most fundamental axiomatic identities such as the inverse relationship between plus and minus, and between multiplication and division, most of which can be matched with the human-predefined rules in Halide. Second and more importantly, HISS can also uncover some more complicated rules that have no matches in Halide, nor could be equivalently derived from any composition of the rules in Halide. A close inspection into these rules reveals that these rules, despite their complexity, are still very fundamental, and therefore the failure to include these rules is expected to impact the simplification performance. In addition to these fundamental rules, HISS is also able to find some involved but interesting rules, which are listed in table 4 and discussed in appendix B.2.

**Example Simplification Traces.** To illustrate how the completeness of the rules can impact on the simplification performance, table 2 compares the simplification traces of Halide and HISS for the following expression

$$(((((144 - (v0*72))/2)*2) + 4) \leq ((150 - (v0*72))/4)*4) \tag{10}$$

For Halide, each step represents the process of applying one Halide predefined rule to simplify the expression. For HISS, each step represents one simplification iteration. As each step, the subtree that is chosen for simplification for the next step is marked with box, unless the entire expression is chosen. As can be seen, HISS follows some reasonable steps to trim the constants and eliminates v0, but Halide gets stuck after some trivial constant reduction. The reason for this failure is there are no such rules as $(((c0 - (v0 * c1))/c2) * c2 \mapsto c0 - (v0 * c1)$ or $x + y \mapsto y + x$ or $(x + y) * z \mapsto x * z + y * z$ in the Halide ruleset. Of course, one can fix this problem one time by appending the aforementioned rules to the ruleset, but this does not fundamentally solve the problem because one would never be able to exhaustively list all the possible rules needed for simplification in the ruleset.

**Why Subtree Selector Matters.** Table 2 also illustrates why the subtree selector is an integral part of HISS. In many reduction steps, only a subexpression is simplified (as boxed). Without the subtree selector, the decoder of HISS would have to process the entire expression only to simplify the subexpression, which makes it hard to effectively learn via reinforcement learning. For more analysis on the effect of the subtree selector, please refer to appendix C.

## 6 CONCLUSIONS

We presented HISS as a symbolic expression simplification algorithm that is independent of human knowledge. We demonstrated that removing the dependence on humans is advantageous for this task

because machines can autonomously figure out rewrite rules that humans fail to discover, and thus achieve comparably well simplification results. We also showed that we are one step closer to finding an equivalence-preserving embedding for symbolic expressions. Although HISS has achieved promising results, there is still much room for improvement. Although HISS has adopted several techniques to reduce the complexity of the search space, learning simplification rules on very long expressions is still challenging, which calls for the exploration of more efficient reinforcement learning algorithms as a future research direction.

## 7 ACKNOWLEDGEMENT

We thank the anonymous reviewers for their valuable feedback. This paper is supported by NSF grant 1829525.

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

# A EXPERIMENT SETUP

## A.1 TRAVERSE EQUIVALENCE DATASET

Table 3 shows the vocabulary of Halide syntax, which is used to construct the traverse equivalence dataset. As there are 14 binary operators, 1 unary operator, and 1 ternary operator, the total number of different expressions of depth two would be over 50k, and the number of different expressions of depth three is over 1.25e+14. However, the rule of thumb is that the simplification is most possible when there are some variables repeatedly appear in the expression, for example, $(x + y) - x \mapsto y$ while $(x + z) + y$ could not be simplified. Therefore, we constraint the enumeration of expression to only 16 operators, 5 constants, and the first three variables (v0, v1, v2). As a result, roughly 3 billion expressions are enumerated.

To check whether an expression could be simplified in this plethora of possible candidates, we adopted the FingerPrint method (Jia et al., 2019). The idea is that equivalent expressions should produce the same output under the same set of input. Initially, $n$ sets of random value assignment to all variables are generated, and the fingerprint of an expression is defined as the tuple of the corresponding $n$ output given the assignment. According to the fingerprint, expressions with exactly $n$ ($n = 4$ in our case) same results are grouped into a shard. It can be implied that any equivalent expressions must be in the same shard. Thus, the shards could be processed in parallel. It is highly possible that the initial assignment would lead to an extremely large shard. Therefore, when processing each shard, we repetitively compute new fingerprints until the shard breaks into small shards containing fewer than 5,000 expressions. Then, we use probabilistic testing on each pair of expressions from the same shard to determine whether the expression pair is equivalent. For each expression pair that is equivalent, the longer expression is labeled as reducible. After all the pairs are tested, all the expressions that are labeled as reducible are retained, and the rest are removed.

Further, it is obvious that if an expression could be simplified, the longer expression containing it could also be simplified. So if an expression contains a subexpression that could be simplified, then this expression does not represent the minimal sequence of this simplification and is removed from the dataset as well. After the FingerPrint method and the minimality checking, we obtain approximately 20,000 equivalent expression pairs, from which 1,500 samples are randomly sampled and further split into training, testing, and validation set with a ratio of 6:2:2.

## A.2 HALIDE DATASET

The Halide dataset (Chen & Tian, 2018) contains around 10,000 training sequences, 1,000 testing sequences, and 1,000 validation sequences, generated and split randomly. The number of words for each sequence ranges from 6 to 100, averaged at 58. The expressions generated contain many constants beyond {0, 1, 2, True, False}, and the constant is renamed to constant symbols shown in table 3, and the same constant value is renamed to the same constant symbol. There are at most 14 constant symbols and at most 15 variables in a single expression.

## A.3 HYPERPARAMETER SETTING AND DETERMINATION

The input to the network is one-hot encoded sequences where the vocabulary size is 50, then the input is encoded by a single fully connected layer with output size 32. The hidden units of LSTM are set to 64 for both encoder and decoder, as a common setting adopted in many previous works (Liu & Lee, 2017), and the number of layers is 1. The output size of the encoder is 64, and the output size of the decoder is equal to the vocabulary size (50). The subtree selector consists of two feed-forward layers with output sizes of 128 and 1 respectively. The model is trained with the ADAM optimizer with a learning rate of 1e-3. Rather than tuned on the validation set, this hyperparameter setting is determined by following the common setting in previous works ((Liu & Lee, 2017)) as well as referencing to the Halide vocabulary size. The same hyperparameter setting has been applied throughout this research project. Finetuning the hyperparameters is expected to have a minor effect on the performance as compared to refining our major algorithm design, *e.g.* introducing subtree selector and $\ell_2$ embedding regularizer, which will be discussed in further details in the ablation studies in appendix C.

Table 3: The vocabulary of the symbols and operators that are used to construct the traverse equivalence dataset.

| Type | Symbols |
|------|---------|
| Operators | min, max, $\geq$, $\leq$, $<$, $>$, ==, !, $\neq$, select, +, -, *, /, &&, \|\| |
| Constants | 0, 1, 2, True, False |
| Variables | v0, v1, ..., v14 |
| Constant symbols | c0, c1, ... c13 |

The penalty of not getting an equivalent expression, $\beta$ (as in Eq. (8)), is set to 0.1. This is motivated by our observation that in the Monte Carlo sampling results the ratio of equivalent expressions over nonequivalent expressions is roughly 10:1 on a randomly initialized model. Therefore, by matching $\beta$ to this ratio we can balance the reward and penalty and achieve an average reward of around zero, at least for the initial iterations, which is shown to contribute to a more stable REINFORCE training. However, please be reminded that $\beta$ is not a crucial parameter because the baseline removal process in REINFORCE would automatically balance the reward and penalty. A good choice of $\beta$ would mostly only benefit the onset of the training when the baseline estimation is not yet accurate. The weight to embedding similarity loss term is set to 0.1.

## A.4 TRAINING AND INFERENCE

As mentioned, the training for section 5.1 involves only the first stage in the curriculum learning, whereas the training for section 5.2 involves both stages in the curriculum learning. Below are the details regarding the training and inference schemes.

**Training Iterations** For the stage-one training in both experiments, we apply an identical setting, where the encoder-decoder model is trained for 10 epochs (90,000 steps). The model with the highest hit rate on the validation set is selected for evaluation in section 5.1 as well as for the stage-two training of the curriculum. The stage-two training takes two weeks to train the HISS for full simplification pipeline on RTX 2080 for 40 epochs (400k steps).

**Beam Search** The beam search algorithm is performed as follows. In the beginning, the top $k$ choices of the root node with the highest probability are decoded. Then for each choice of root node value, the next step would be to decode the top $k$ choices of each child node of the root node. Since the decoding processes of different child nodes of the same parent node are independent, we then perform the Cartesian product of all the $k$ choices for each node at the current step, and preserve the top $k$ combinations with the highest probability. Repeating this way, at step $t$, beam search decodes $k$ highest probability tree up to depth $t$. Finally, top $s$ ($s < k$) expressions are backtracked and used to estimate the expected reward of the model. The probability of beam search decoded expression is re-normalized according to Bunel et al. (2018). In all the experiments, we set beam size $k = 20$ and $s = 20$.

**Constant Folding** Since the Halide dataset contains a large number of constant values, we apply a technique called constant folding to both stage-two training and inference of HISS as well as to all other baselines. Specifically, once the expression is rewritten by the neural network in the symbolic domain, it will be checked if all the leaf nodes in the subtree are constant. If so, the expression is executed and replaced by a new single node with the execution result. Constant folding is applied in both training and inference.

## B ADDITIONAL EXPERIMENT RESULTS

### B.1 ATTENTION VISUALIZATION

To understand the attention mechanism, we visualize the attention to the input sequence shown in figure 4. We find that when decoding an operator, the attention tends to be flat (with a few exceptions in the right two figures), because it needs to understand the overall logic. This is different from machine translation or summarization, where the output attention of a single word is usually focused on several input tokens. On the other hand, when decoding a variable, the model attends sharply to the corresponding variable in the input.

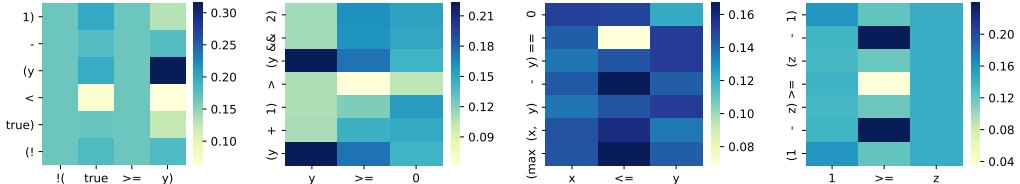

Figure 4: Attention weights on the input sequences for each token decoded. The x-axis shows the output sequence, and the y-axis shows the input sequence. The input sequence is encoded from leaves to root and the output sequence is decoded from root to leaves. Tokens are re-arranged in the natural order for better visualization.

Table 4: Involved simplification rules discovered by HISS.

| Input | Output |
|---|---|
| $(!\text{true}) < (y - 1)$ | $!(\text{true} \geq y)$ |
| $(y - \text{true}) \| (y \geq \text{true})$ | true |
| $\max(z, 1) \geq (\text{false}\|x)$ | true |
| $(1 - z) \geq (z - 1)$ | $1 \geq z$ |
| $\min(x, \text{true})\&\&(z/2)$ | $x\&\&z$ |
| $1 - (1 < x)$ | $1 \geq x$ |
| $\text{select}(z, z, \text{true}) == \text{select}(z, \text{false}, \text{true})$ | $!z$ |
| $(x * y)\&\&\text{true}$ | $y\&\&x$ |
| $(y + 1) > (y\&\&2)$ | $y \geq 0$ |

Table 5: Mean and standard deviation of the performance metrics among random initialization. The experiment setting is the same as in section 5.1.

|  | Sampling Parameter | | | |
|---|---|---|---|---|
|  | 5 | 10 | 15 | 20 |
| **Hit Rate** | $0.391 \pm 0.013$ | $0.391 \pm 0.009$ | $0.406 \pm 0.014$ | $0.406 \pm 0.014$ |
| **Expression Length Reduction** | $2.250 \pm 0.172$ | $2.250 \pm 0.140$ | $2.578 \pm 0.242$ | $2.578 \pm 0.242$ |
| **Tree Size Reduction** | $1.078 \pm 0.077$ | $1.078 \pm 0.069$ | $1.219 \pm 0.114$ | $1.219 \pm 0.114$ |

## B.2 EXAMPLES OF INVOLVED SIMPLIFICATION RULES

In order to further appreciate the ability of HISS in finding equivalence, in addition to the rules listed is table 1, we list some simplification rules discovered by HISS on randomly generated expressions that are more involved in table 4. In fact, it takes the authors quite a while to figure out the equivalence. These rules are hardly useful in practice because no humans will code in this way, but it is a vivid illustration of the advantage of HISS in finding powerful simplifications beyond human knowledge.

## B.3 ROBUSTNESS AGAINST RANDOM INITIALIZATION

To assess if the performance of HISS is robust to random model initialization, we perform the same experiment in section 5.1 eight times with different random initialization and compute the mean and standard deviation of the three metrics, which are shown in table 5. Compared to the absolute value of the mean, the standard deviation is very small, which shows that the random initialization has a minor influence on the performance of HISS.

## C ABLATION STUDIES

In this section, we introduce a set of ablation studies that investigate the significance of the major components of HISS in terms of contribution to performance. The major components of interest are the subtree embedding similarity loss as in Eq. (9), the subtree selector as introduced in section 3.3, and the tree-LSTM encoder-decoder architecture as in sections 3.2 and 3.4.

As an overview, figure 5 compares the performance between HISS and the following variants of HISS on the Halide dataset.

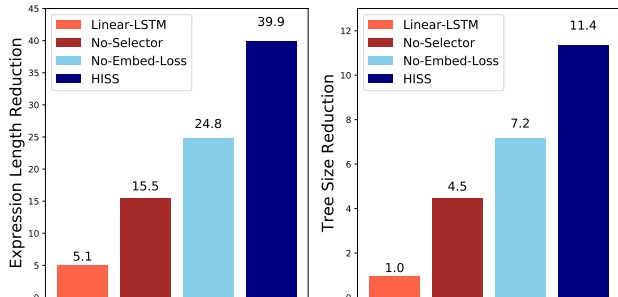

Figure 5: Performance of different variants of HISS on the Halide dataset.

- *No-Embed-Loss:* The original HISS trained without the subtree embedding similarity loss.

- *No-Selector:* The original HISS without the subtree selector.

- *Linear-LSTM:* The Tree-LSTM structure is removed and a simple linear-LSTM is used instead. The embedding loss as well as the subtree selector are no longer applicable and therefore also removed.

Other than the variations aforementioned, all the experiment settings are identical to the experiment in section 5.2. As can be seen in figure 5, without training with the embedding similarity loss or the subtree selector, the performance is significantly compromised. Furthermore, removing the tree-LSTM structure leads to almost a complete failure. The subsequent subsections further investigate why each of these modules has such a significant impact on the performance respectively.

## C.1 SUBTREE EMBEDDING SIMILARITY

We would like to investigate the specific effects that introducing the embedding similarity loss brings.

The direct goal of the embedding similarity loss is to better cluster the embeddings whose corresponding expressions are equivalent. Therefore, we would like to first check if this direct goal is achieved. To evaluate this, in the experiment described in section 5.1, we select the six most-populated subsets of equivalent expressions in the test set and evaluate the similarity of the embeddings computed by HISS in two ways. First, the embeddings are further projected to two-dimensional space using t-SNE (Maaten & Hinton, 2008), which forms a scatter plot as in Figure 6(a-1). The points corresponding to equivalent expressions are shown in the same color. As can be seen, the embeddings equivalent expressions are highly clustered. Notice that this result is on the low-dimensional projection of the embedding. To better evaluate their similarity in the original space, we compare their inter- and intra-subset distances. The inter-/intra-subset distance of a subset is defined as the Euclidean distance between the centroid of the subset and the samples outside/within the subset. Figure 6(a-2) illustrates the box plot of these distances. As shown, there is a significant difference between intra- and inter-subset distances. Except for the first subset, the quartile intervals are well separated.

Figure 6(b-1) and (b-2) show the same plots on the No-Embed-Loss model, *i.e.* the HISS variant that is trained without the embedding similarity loss. As can be seen, the scatter points are apparently less well-clustered, and the difference between inter- and intra-subset distances, although still exists, is smaller. Therefore, we can draw two conclusions. First, even without the embedding similarity loss, HISS is still able to learn embeddings that are somewhat clustered according to equivalence, which shows the goal of finding equivalent expressions roughly aligns with the need to cluster the embedding based on the expression equivalence. Second, the embedding similarity loss can improve embedding clustering.

However, what we are more interested in is why an improved clustering of the embeddings would lead to a significant performance gain. To answer this question, figure 7 compares the average reward as a function of training epoch of HISS and its variant without the embedding similarity loss, which gives us some very interesting insights. Notice that despite their initial difference, which is due to the random initialization, both reward curves reach the same plateau at around epoch seven, where they both become stagnant for a while. However, with the help of embedding similarity loss, HISS

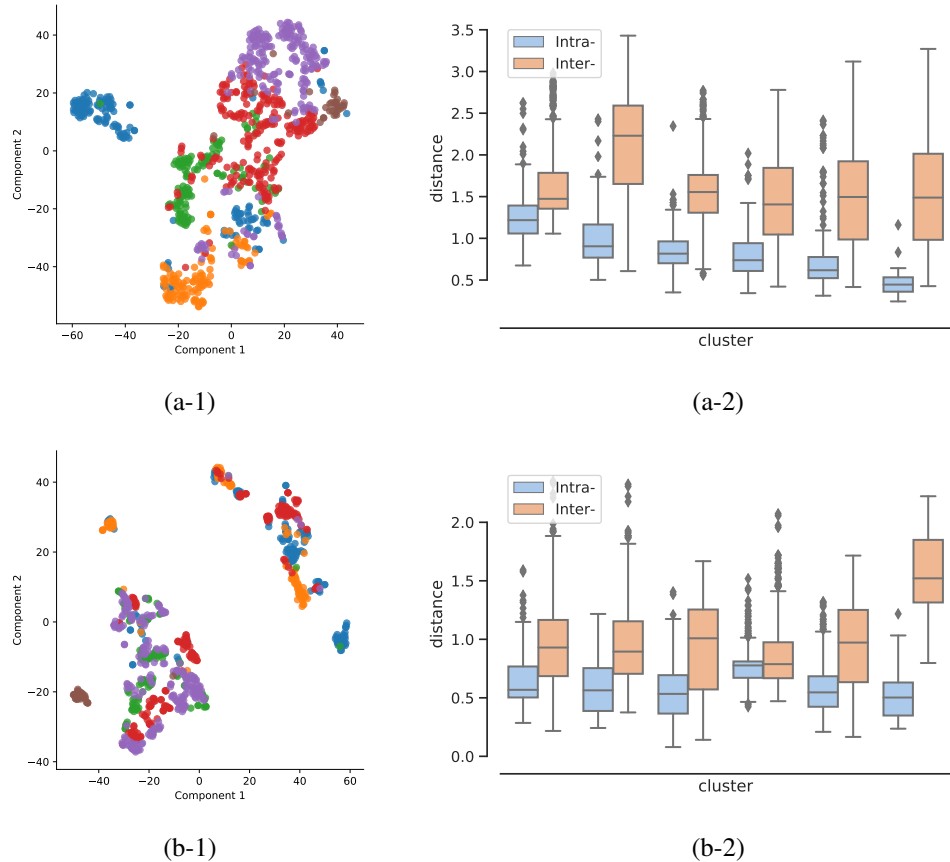

(a-1)          (a-2)

(b-1)          (b-2)

Figure 6: Evaluation of similarity of the embeddings of equivalent expressions. The upper plots ((a-1) and (a-2)) are evaluated on the original HISS model. The lower plots ((b-1) and (b-2)) are evaluated on the No-Embed-Loss model. Left plots ((a-1) and (b-1)) are scatter plots of embeddings projected onto two-dimensional space using t-SNE. Points corresponding to equivalent expressions are shown in the same color. Right plots ((a-2) and (b-2)) are box plots of intra- (blue) and inter-subset (orange) distances of the embeddings trained with $\ell_2$ loss. The bars in the box represent 25%, 50% and 75% quartile values. The line intervals denote the 1.5 interquartile range (IQR) beyond the quartile values. The dots represent the extreme values.

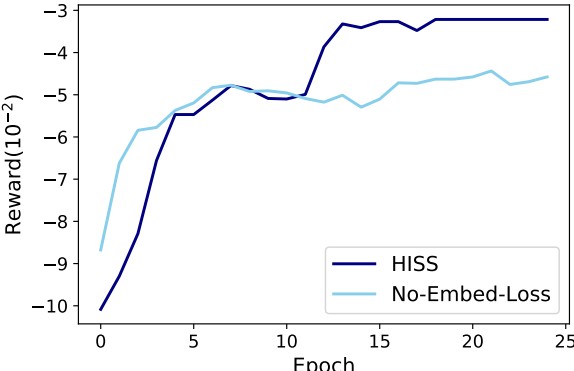

Figure 7: Expected reward of HISS with the embedding similarity loss (HISS) and without the embedding similarity loss (No-Embed-Loss)

Table 6: Simplification traces of HISS with and without the subtree selector on Eq. (11). The subexpressions selected by the subtree selector are boxed, unless the entire expression is chosen.

| Step | HISS with the Subtree Selector |
|------|--------------------------------|
| 0 | ((v1+v2)-7)≤((( $\boxed{((max(v1,16)+18)/8)*8)}$ +(v1+v2))-27) |
| 1 | ((v1+v2)-7)≤(((max(v1,16)+18)+(v1+v2))-27) |
| 2 | (v1+v2)≤(((max(v1,16)+18)+(v1+v2))-20) |
| **Step** | **HISS without the Subtree Selector** |
| 0 | ((v1+v2)-7)≤(((((max(v1,16)+18)/8)*8)+(v1+v2))-27) |
| 1 | (v1+v2)≤(((((max(v1,16)+18)/8)*8)+(v1+v2))-20) |

Table 7: Output sequences of Linear-LSTM trained on Traverse Equivalence dataset

| Linear-LSTM with only equivalence reward | |
|------|------|
| **Input** | **Output** |
| max(v1 + v0, v2 + v0) | [',', 'c8', '2', '2', '1', 'v9', ',', 'c8', '2', '2'] |
| min(min(v2, v0 + v1), v0 + v3) | [',', 'c9', 'c9', 'c9', 'c9', '2', '2', '1', ',', 'c9'] |
| (v0 &&(v2||v1))||v1 | [',', 'c8', '2', '2', '1', ',', 'c9', 'c9', 'c9', '2'] |
| **Linear-LSTM trained with equivalence and valid expression reward** | |
| **Input** | **Output** |
| min(v2, v0 + v1) - v1 | ['v1'] |
| v1||(v0||v1) | ['c0'] |
| v0 - min(v2, v0 + v1) | ['2', '≥', 'v0'] |

is finally able to escape from the plateau and reach a higher reward level, whereas the one without the embedding similarity loss gets trapped in the plateau. This result suggests that the embedding similarity loss provides an extra training signal to address the convergence issue of REINFORCE.

## C.2 SUBTREE SELECTOR

We have already demonstrated the inner-workings of the subtree selector in section 5.3 and shown in figure 5 that subtree selector is indispensable for the good performance of HISS. Here we would like to intuitively explain why this is the case. Table 6 compares the simplification traces of HISS with and without the subtree selector on the following expression

$$((v1+v2)-7)\leq(((((max(v1,16)+18)/8)*8)+(v1+v2))-27) \tag{11}$$

As can be seen, HISS with the subtree selector can first offset the '*8' and '/8' terms in the subexpression ((max(v1,16)+18)/8)*8), before it applies the cancellation rule to merge the constants '-7' and '-27'. On the other hand, HISS without the subtree selector is unable to identify the reducible subexpression, and so it only applies the cancellation rule. This result shows that the reason why the subtree selector helps is that it can thoroughly check on each subexpression, and therefore contributes to a better simplification. Without the subtree selector, the algorithm is prone to overlook some small subexpressions that are reducible.

## C.3 TREE LSTM V.S. LINEAR LSTM

We can see the conspicuous disadvantage of the Linear-LSTM model from figure 5. To illustrate the fundamental issue with the Linear-LSTM model, we sampled some output of the model trained on the Traverse Equivalence dataset, listed in table 7. The model was trained with REINFORCE for 200k steps, with two different reward settings: 1) the equivalence reward as in Eq. 8; 2) the equivalence reward plus a valid expression reward, which is an additional reward of 0.1 if the decoded sequence is syntactically correct.

As can be seen in table 7, when trained with only the equivalence reward, the linear decoder is unable to decode even a valid expression, not to mention generating an equivalent expression to the input. So the linear decoder could hardly converge and learn useful information from reward, which was almost always a negative constant. As a remedy, we can add a valid expression reward to guide

the linear LSTM to generate valid expressions. However, as can be observed in table 7, under this additional reward, the model would overfit this reward by only generating short valid expressions, which are not equivalent to the input. This is because short expressions contain only one variable or constant have a higher generation probability than longer valid expressions. During the training, whenever the model happens to generate a single expression, it was rewarded a positive valid expression reward. So the behavior of generating only a single constant or variable is strengthened. Still, the probability that a random variable was equivalent to the input expression, is small, and therefore the model can hardly be guided by equivalence reward, but focuses only on generating short but valid expressions.

This experiment demonstrates the advantage of using the tree LSTM, which is guaranteed to generate syntactically correct, and which has a much higher probability of hitting an equivalent expression. Therefore, the tree LSTM can be much better guided by the equivalence reward.

