# OpenReview forum: "Deep Symbolic Superoptimization Without Human Knowledge"
_ICLR.cc/2020/Conference — Accept (Poster)_

### Official Review · AnonReviewer3 · 2019-10-22
**Official Blind Review #3**

**Rating:** 6

**Review:**

This paper provides a novel approach to the problem of simplifying symbolic expressions without relying on human input and information. To achieve this, they apply a REINFORCE framework with a reward function involving the number of symbols in the final output together with a probabilistic testing scheme to determine equivalence. The model itself consists of a tree-LSTM-based encoder-decoder module with attention, together with a sub tree selector. Their main contribution is that this framework works entirely independent of human-labelled data. In their experiments, they show that this deep learning approach outperforms their provided human-independent baselines, while sharing similar performance with human-dependent ones.
Overall, the work in this paper has the potential to be a contribution to ICLR but lacks completeness and clarity. A number of experimental details are missing, making it difficult to understand the setup under which results were obtained. Moreover, the paper does not seem to have been revised with many grammatical issues that make it hard to read.
The following are major issues within the paper and should be addressed:
•	The paper does not mention the amount of compute given to their model, nor the amount of time taken to train. As the REINFORCE framework is generally quite computation-heavy, these are significant details. Without assessing the amount of compute and time allotted for training HISS, the comparisons to previous baselines lose a fair amount of meaning. The paper alludes to processes being ‘extremely time consuming’, but then does not provide any numbers.
•	They do not mention the data used to train the model weights. In the comparisons sections, some details on datasets are given, but these seem to refer to data for inference.
•	There are many grammatical errors that likely could have been detected with 	 revision. A handful of such errors would not affect the score, but they are so numerous as to make the paper much more difficult to understand.
Additionally, these are comments that slightly detract from the quality of the paper:
•	It’s unclear what to glean from Section 5.1, as the dataset and baselines seem to be fairly trivial. If their claim is to have the first nontrivial human-independent approach to simplifying symbolic expressions, there is no need to compare to baselines that can only handle small expressions.
•	Sections 5.3 and 5.4 contribute little to the paper. For 5.3, the model was trained to embed equivalent expressions close together, using L2 regularization. It is therefore unsurprising that equivalent expressions are then closer together than non-equivalent ones. The paper also does not provide a comparison to the method without this regularization, and so it’s unclear if this embedding similarity helps in any way.  For 5.4, the section is extremely short and contains very little content. Moreover, just as many of the variables in their provided examples oppose their conjectures as support them.
•	The most interesting figure provided is the rewrite rules discovered by the model. It would be even better if an additional column containing the rules discovered by Halide (the main baseline) were provided.
Overall, in my understanding, the primary point in favor of the paper is in being the first nontrivial human-independent approach to simplifying symbolic expression. That said, this is not my area of expertise, so I cannot judge novelty or importance as well as other reviewers.


**Experience Assessment:**

I do not know much about this area.

**Review Assessment: Checking Correctness Of Derivations And Theory:**

I carefully checked the derivations and theory.

**Review Assessment: Checking Correctness Of Experiments:**

I assessed the sensibility of the experiments.

**Review Assessment: Thoroughness In Paper Reading:**

I read the paper thoroughly.

---

> ### Author Response · Authors · 2019-11-14
> **Response to Review #3**
>
> We have updated our paper. Please refer to the thread “Revised Paper Uploaded” for a summary of major updates in our revision. Following is a detailed response to Review #3 comments.
>
> 1. Training details.
> We add Appendix A.4 to describe training computation complexity. The most time-consuming portion of training is stage-2, which requires two weeks to converge, using a single RTX 2080.
>
> 2. Datasets.
> Our revision Section 5.1 and Appendix A.1 & A.2 provide a detailed description of datasets. We use two datasets: the “traverse equivalence dataset” is used for stage-I training; a benchmark dataset “Halide dataset” is used by the stage-II training. The former dataset mainly contains short expressions under a depth of four. Section 5.2 experiment is trained and tested on the traverse equivalence dataset; Section 5.3 experiment employs (a) the traverse equivalence dataset for stage-1 training and (b) the Halide dataset for stage-2 training and testing.
>
> 3. What to glean from Section 5.1 (Now Section 5.2).
> Although the expressions to be evaluated are below depth 4, they can be very complicated expressions already. For example, min(min(v2, v0 + v1), v0 + v3)  is a depth-4 expression, and is included in the dataset. Since HISS applies a beam search algorithm, it is useful to demonstrate how HISS differs from conventional search algorithms and that the simplification problem cannot be trivially solved by these search algorithms.
>
> 4. Sections 5.3 and 5.4.
> We move the original Sections 5.3 and 5.4 to the appendix. We keep them in the appendix to provide insights into how HISS works.
>
> 5. Method without embedding similarity loss.
> Our revision Appendix C provides a set of ablation studies. Specifically, Figure 5 shows how significantly performance drops if the subtree embedding similarity loss is removed. Appendix C.1 discusses why the embedding similarity loss is important. In short, this loss will provide a more concrete training signal to assist convergence, which is otherwise difficult due to REINFORCE. In addition to the embedding similarity loss, the ablation studies also investigate the significance of the subtree selector and the tree-LSTM architecture.
>
> 6. Rewrite rules comparison.
> Section 5.4 (especially Tables 1 and 2) of our revision includes our new results, which directly compare HISS rewrite rules with Halide rewrite rules. Our results show that HISS rules are more comprehensive. We further compare the simplification traces of HISS and Halide, demonstrating how the comprehensiveness of the rewrite rules is translated to HISS’s performance benefit.

---

> > ### Comment · AnonReviewer3 · 2019-11-15
> > **Re: Response to Review #3**
> >
> > The revisions strengthen the paper, and I have updated my score accordingly.

---

### Official Review · AnonReviewer2 · 2019-10-23
**Official Blind Review #2**

**Rating:** 3

**Review:**

The authors present a framework for symbolic superoptimization using methods from deep learning. A deep learning approach operating on the expression tree structures is proposed based on a combination of subtree embeddings, LSTM RNN structures, and an attention mechanism.

The approach avoids the exploitation of human-generated equivalence pairs thus avoiding human interaction and corresponding bias. Instead, the approach is trained using random generated data. It remains somewhat unclear how the corresponding random data generation influences general applicability w.r.t. other tasks, as the authors apply constraints on the generation process for complexity reasons. A corresponding discussion would be valuable here.

In Secs. 3 & 4, the authors present their specific modeling and learning approach. However, they do not report on modeling or learning alternatives. It would be interesting for the audience to understand, how the authors reached these specific choices, and how (some of) these choice influence performance and learning stability. For example, in Sec. 4.1, an additional loss term is introduced to further support the learning of embeddings. However, it might interesting to see comparative results quantitatively investigating the effect of this additional loss term. Also, as far as I can see, no information on the choice of hyperparameters (e.g. LSTM dimensions) are provided or analyzed w.r.t. their effect on the performance of the proposed approach.

**Experience Assessment:**

I do not know much about this area.

**Review Assessment: Checking Correctness Of Derivations And Theory:**

I assessed the sensibility of the derivations and theory.

**Review Assessment: Checking Correctness Of Experiments:**

I assessed the sensibility of the experiments.

**Review Assessment: Thoroughness In Paper Reading:**

I read the paper at least twice and used my best judgement in assessing the paper.

---

> ### Author Response · Authors · 2019-11-14
> **Response to Review #2**
>
> We have updated our paper. Please refer to the thread “Revised Paper Uploaded” for a summary of major updates in our revision. Following is a detailed response to Review #2 comments.
>
> 1) Ablation studies.
> Our revised Appendix C introduces a set of ablation studies. Specifically, figure 5 shows that the performance drops significantly if the subtree embedding similarity loss is removed. In addition, appendix C.1 explores why the embedding similarity loss is important. In short, this loss will provide a more concrete training signal to assist convergence, which is otherwise difficult due to REINFORCE. In addition to the embedding similarity loss, the ablation studies also investigate the significance of the subtree selector and the tree-LSTM architecture. For more details, please kindly refer to appendix C.
>
> 2) Hyperparameter settings.
> Our hyperparameters, including the LSTM dimensions, follow the common setting in previous works, without any explicit tuning. Our revised Appendix A.3 describes our hyperparameter setting and our decision making in detail.
>
> 3) How data generation affects performance.
> Our revised Section 5.1 and Appendix A.1 & A.2 provide a detailed description of datasets preparation and use. We use two datasets: (a) the “traverse equivalence dataset” (mentioned by Review#3) is only used for stage-I training; (b) the benchmark dataset “Halide dataset” is used by the stage-II training. The former dataset mainly contains short expressions under a depth of four; the goal is to assist the stage-II training to be performed on the latter dataset. Therefore, It does not affect the performance that the traverse space is reduced when generating the dataset for complexity reasons. Section 5.1 and Appendix A.1 & A.2 describes more details about the datasets.

---

### Official Review · AnonReviewer1 · 2019-10-28
**Official Blind Review #1**

**Rating:** 6

**Review:**

This paper presents a method for symbolic superoptimization — the task of simplifying equations into equivalent expressions. The main goal is to design a method that does not rely on human input in defining equivalence classes, which should improve scalability of the simplification method to a larger set of expressions. The solution uses a reinforcement learning method for training a neural model that transforms an equation tree into a simpler but equivalent one. The model consists of (i) a tree encoder, a recursive LSTM that operates over the input equation tree, (ii) a sub-tree selector, a probability distribution over the nodes in the input equation tree, and (iii) a tree decoder, a two layer LSTM that includes a tree layer and a symbol generation layer. The RL reward uses an existing method for determining soft equivalence between the output tree and the input tree along with a positive score for compressing.

The main strengths of the paper are that (i) it targets the scalability problem in simplifying arithmetic expressions by reducing the amount of human effort involved in the process, (ii) it provides some evaluation comparing against methods that use pre-defined rules for transforming equations, (iii) it provides a baseline method against which future scalable models can be compared against.

The following are the main concerns I have with the paper

1) The model description should include more details about the design choices.
For instance, the specific reward function, the central component of the model, is left rather under-discussed. The function form is sensible but why does the negative case use -0.1 as the reward scaling constant? Why not some other number? How is this tuned?

2) As far as I can see there is no real ablation analysis that shows which of the components are actually useful. Is the sub-tree selector necessary? Is the curriculum training necessary? How much does the sub-tree embedding similarity loss contribute to the results? Even if each of these actually add value it will be useful to know how much. What about other design choices? If we trained a direct seq2seq model with linearized expressions instead of the tree structured inputs, would it work just as well? These are empirical questions that need to be answered to justify that the proposed model indeed is useful.

3) The experimental details are sparse. In particular, there is no mention of how hyper-parameters of the proposed method are tuned. Are the performance numbers averages over a set of random seeds or is it simply the best performing number that has been reported? This is especially troublesome for a RL based model.

Overall the paper presents a particular model and strategy for training but lacks appropriate experimentation to establish their utility.



**Experience Assessment:**

I do not know much about this area.

**Review Assessment: Checking Correctness Of Derivations And Theory:**

I did not assess the derivations or theory.

**Review Assessment: Checking Correctness Of Experiments:**

I assessed the sensibility of the experiments.

**Review Assessment: Thoroughness In Paper Reading:**

I read the paper at least twice and used my best judgement in assessing the paper.

---

> ### Author Response · Authors · 2019-11-14
> **Response to Review #1**
>
> We have updated our paper. Please refer to the thread “Revised Paper Uploaded” for a summary of major updates in our revision. Following is a detailed response to Review #1 comments.
>
> 1) Details of design choices.
> Our revision Appendix A.3 describes details of our design choices and the rationale behind making them. These choices were based on reference to previous common settings and our reasoning, rather than tuning. For example, regarding why using -0.1 as the reward scaling constant: we observed that for the Monte Carlo sampling, the probability of hitting equivalent expressions is around 10%; therefore, we set the penalty to be 0.1 to balance the reward and penalty, such that the average reward is close to 0. Also note that this constant is not a crucial hyperparameter, because the baseline removal procedure of REINFORCE will take the place to balance the reward and penalty during training. Appendix A.3 has more details about various design choices.
>
> 2) Ablation studies.
> Our revision Appendix C presents our ablation studies. Figure 5 shows a quantitative breakdown of the contributions of each design component. In particular, the performance significantly drops if we remove either the subtree embedding similarity loss or the subtree selector. If a linear seq2seq model is used instead of the tree-LSTM, the algorithm almost completely fails. The following are more detailed explanations.
>
> - The contribution of the embedding similarity loss is to improve training convergence (discussed and evaluated in Appendix C.1). Without the concrete training signal provided by this loss, the training can be stuck at a poor local optimum due to REINFORCE.
>
> - The contribution of the subtree selector is to ensure a broad coverage of simplifiable subexpressions, by helping pinpoint the subtrees that can be simplified (discussed and evaluated in Appendix C.2 and Section 5.4). Without the subtree selector, the algorithm will tend to overlook certain small subexpressions that can be simplified.
>
> - Appendix C.3 explains why tree-LSTM is much more effective than linear seq2seq. In short, a linear seq2seq model will have a hard time learning the correct syntax. With seq2seq, the probability of hitting a syntactically correct expression is already low, not to mention the probability of hitting any equivalent expressions.
>
> - The contribution of curriculum learning is to ensure a sufficiently high probability of hitting the equivalent expressions, in order to provide a meaningful training signal in RL.
>
> 3) Hyperparameters.
> Our revision Appendix A.3 describes our hyperparameter setting and our decision making in detail. The original results reported are from a single run of the inference. To address the review concern about the potential deviation in results if more runs were performed, our revision performs a multiple-run evaluation on the experiment described in Section 5.2. The corresponding results are reported in Appendix B.3, demonstrating consistently promising results.

---

### Author Response · Authors · 2019-11-14
**Revised Paper Uploaded**

We thank the reviewers for their valuable feedback. Major concerns include the need for a more detailed description of our design choices, more explanation on experiment results, additional experiments, and ablation studies. We uploaded a revised version according to review comments.

Following is a summary of our major updates (less major updates are described in response to each reviewer).

1. We add a detailed description of our experiments, including a clearer description of our dataset in Section 5.1, and additional details in Appendix A.

2. We perform a set of extensive ablation studies on key model designs in Appendix C, including embedding similarity loss (C.1), subtree selector (C.2), and tree-LSTM structure (C.3). We not only compare the performance with and without these designs, but also provide intuitive and empirical explanations of why they work.

3. We add new experiment results, including a back-to-back comparison of the simplification processes of HISS and Halide (Section 5.4), and results with various initializations (Appendix B.3). The highlight of the new experiments is in Tables 1 and 2. Due to page limitation, the original Sections 5.3 to 5.5 are moved to the Appendix B and C.1.

4. We update the results in the experiment introduced in Section 5.3. Our original experiment did not train the model until convergence due to time constraints and the result reported was produced by the half-trained HISS. Now the result is updated with that produced by the fully-trained HISS. Yet, our observations and conclusion drawn from the updated result are in line with our original ones.

---

### Decision · Program_Chairs · 2019-12-19

**Decision:**

Accept (Poster)

**Comment:**

This work introduces a neural architecture and corresponding method for simplifying symbolic equations, which can be trained without requiring human input. This is an area somewhat outside most of our expertise, but the general consensus is that the paper is interesting and is an advance. The reviewer's concerns have been mostly resolved by the rebuttal, so I am recommending an accept.